# Melatonin Promotes the Proliferation of Chicken Sertoli Cells by Activating the ERK/Inhibin Alpha Subunit Signaling Pathway

**DOI:** 10.3390/molecules25051230

**Published:** 2020-03-09

**Authors:** Ke Xu, Jun Wang, Hongyu Liu, Jing Zhao, Wenfa Lu

**Affiliations:** 1Joint Laboratory of Modern Agricultural Technology International Cooperation, Ministry of Education, Jilin Agricultural University, Changchun 130118, Chinajlndlhy0133@163.com (H.L.); 2Key Lab of Animal Production, Product Quality and Security, Ministry of Education, Jilin Agricultural University, Changchun 130118, China; 3Jilin Province Engineering Laboratory for Ruminant Reproductive Biotechnology and Healthy Production, College of Animal Science and Technology, Jilin Agricultural University, Changchun 130118, China

**Keywords:** melatonin, proliferation, inhibin alpha subunit, ERK1/2, Sertoli cells, chicken

## Abstract

Melatonin influences physiological processes such as promoting proliferation and regulating cell development and function, and its effects on chicken Sertoli cells are unknown. Therefore, we investigated the effects of melatonin on cell proliferation and its underlying mechanisms in chicken Sertoli cells. Chicken Sertoli cells were exposed to varying melatonin concentrations (1, 10, 100, and 1000 nM), and the melatonin-induced effects on cell proliferation were measured by Cell Counting Kit 8 (CCK-8), 5-ethynyl-2’-deoxyuridine (EdU), real-time qPCR, and western blotting. We found that 1000 nM melatonin significantly (*p* < 0.05) promoted cell proliferation in chicken Sertoli cells. Furthermore, melatonin significantly (*p* < 0.05) increased the expression of inhibin alpha subunit (INHA), and the silencing of INHA reversed the melatonin-induced effects on Sertoli cell proliferation. We also found that melatonin activates the extracellular-regulated protein kinase (ERK) signaling pathway. To explore the role of the ERK signaling pathway in melatonin-induced cell proliferation, PD98059 (an inhibitor of EKR1/2) was used to pre-treat chicken Sertoli cells. The melatonin-induced proliferation of chicken Sertoli cells was reversed by PD98059, with decreased cell viability, weakened cell proliferation, and down-regulated expression of the proliferating cell nuclear antigen (PCNA), cyclin D1 (CCND1) and INHA. In summary, our results indicate that melatonin promotes the proliferation of chicken Sertoli cells by activating the ERK/inhibin alpha subunit signaling pathway.

## 1. Introduction

Among the seminiferous tubules are Sertoli cells, spermatogonia, spermatocytes, spermatids, and spermatozoon [1], but Sertoli cells are the only somatic cells that play a role in testicular development and function [2]. Sertoli cells regulate spermatogenesis, form a blood-testis barrier, and provide structural, immune, and nutritional support for germ cell development [3,4].

Melatonin (MT) is a neurohormone secreted by the pineal gland at night. It is synthesized by tryptophan under the action of key enzymes arylalkylamine-N-acetyltransferase (AANAT) and N-acetylserotonin-O-methyltransferase (ASMT) [5]. Melatonin is known to play an important role in regulating a variety of physiological processes, including circadian rhythms, sleep, sexual maturity, aging, and neuroendocrine activities [6,7,8]. Melatonin could increase the activity of antioxidant enzymes in the testes, reduce the levels of reactive oxygen species (ROS), nitric oxide (NO) and the abnormal rate of sperm, and alleviate the symptoms of varicocele. These results suggest that melatonin may be a powerful antioxidant that prevents testicular stress-related damage [9]. It has also been demonstrated that melatonin could protect against chemotherapy-induced sperm toxicity by reducing the level of testicular cell apoptosis and increasing cell proliferation [10]. In infertile male testes, melatonin has anti-proliferative, anti-inflammatory and antioxidant effects on local macrophage and mast cell populations [11]. Moreover, melatonin has a pro-apoptotic effect on tumor cells, indicating that melatonin has an anti-tumor effect [12]. In addition, melatonin may increase androgen production in co-cultured Leydig and Sertoli cells by regulating the expression of insulin-like growth factor 1 regulated by melatonin receptor 1 and the inhibition of estrogen synthesis induced by melatonin receptor 1 [13]. There are also reports that melatonin plays an important role in steroid synthesis and male reproduction [14].

In addition, melatonin plays a key role in the regulation of physiological processes such as promoting proliferation [15] and regulating cell development and function [16]. It has been reported that melatonin affects monochromatic light-induced T lymphocyte [15] and B lymphocyte proliferation in chickens [17]. In neonatal rats with hypoxic-ischemic brain damage, melatonin promotes the proliferation of endogenous neural stem cells [18]. Although it was reported that melatonin stimulates the production of glial cell line-derived neurotrophic factor (GDNF) in Sertoli cells by activating the ERK1/2 signaling pathway and promotes the proliferation of goat spermatogonial stem cells [19], the information on the effect and mechanimsm of melatonin on Sertoli cell is still limited.

Inhibin is secreted by Sertoli cells [20,21] and regulated the synthesis and release of follicle-stimulating hormones [22]. Inhibin is composed of an α subunit (INHA) and a βA or βB subunit and is a heterodimeric glandular glycoprotein hormone within the growth factor-β superfamily [23]. Inhibin plays an important role in cell proliferation, and recently study has shown that INHA up-regulates proliferation-related genes and promotes cell proliferation in pig granulosa cells [24] and mice Sertoli cells [25]. Moreover, melatonin promotes the expression of INHA and regulates cell development in bovine Sertoli cells [16]. However, these processes have never been examined in chicken Sertoli cells. We hypothesized that INHA is involved in melatonin-induced proliferation of Sertoli cells. Therefore, we investigated the effects and mechanisms of melatonin on cell proliferation in chicken Sertoli cells. 

## 2. Results

### 2.1. Identification of Sertoli Cells

The oil red O results showed that the lipid droplets were red, the nuclei were blue, and that the bipolar bodies specific to the Sertoli cells were observed in the nucleus (Figure 1A). RT-PCR showed that Sertoli cell-specific genes (SOX9, GATA4) were expressed in the cells (Figure 1B). 

The immunofluorescence results showed that vimentin was positive in the Sertoli cells and the proportion of positive cells reached 94%, which can be used in subsequent experiments (Figure 1C).

### 2.2. Melatonin Promoted the Proliferation of Chicken Sertoli Cells

To study the effects of melatonin on Sertoli cells, we treated the cells with melatonin at 1, 10, 100, and 1000 nM. The results showed that 1000 nM melatonin significantly increased cell viability (Figure 2A) and the proportion of EdU-positive cells (Figure 2B,C) compared with the control group (*p* < 0.05). Next, we examined the expression levels of the proliferating cell nuclear antigen (PCNA) and cyclin D1 (CCND1). The results are shown in Figure 2D–H; 1000 nM melatonin significantly increased the expression levels of PCNA and CCND1 (*p* < 0.05). Based on these results, we used 1000 nM melatonin in the subsequent experiments.

### 2.3. Melatonin Promoted the Expression of INHA in Chicken Sertoli Cells

As shown in Figure 3A,B, the 1000 nM melatonin treatment significantly increased the expression of INHA (*p* < 0.05).

### 2.4. Identification of the Interference Efficiency of INHA siRNA

Sertoli cells were interfered with three INHA siRNAs to inhibit INHA expression. Compared with the negative control group (NC), siRNA1, siRNA2, and siRNA3 significantly reduced the mRNA and protein expression of INHA (*p* < 0.001; Figure 4A,B). These results indicated that siRNA3 can be used in subsequent experiments.

### 2.5. Melatonin Promoted Cell Proliferation by Affecting INHA in Chicken Sertoli Cells

To elucidate the function of INHA in the underlying mechanisms of melatonin-regulated Sertoli cell proliferation, we silenced INHA and examined the effects of melatonin on chicken Sertoli cell proliferation. Silencing INHA reduced cell viability (Figure 5A) and proliferation (Figure 5B,C) compared with the negative control group with melatonin. Silencing INHA also significantly reduced the expression of CCND1 (*p* < 0.01; Figure 5E–G). However, there were no significant differences in PCNA expression (Figure 5D,F,H). In summary, melatonin promotes the proliferation of chicken Sertoli cells by affecting INHA.

### 2.6. Melatonin Promotes Cell Proliferation by Activating the ERK Signaling Pathway and Affecting INHA in Chicken Sertoli Cells

To elucidate the mechanism of melatonin regulation in Sertoli cell proliferation, the expression of key proteins in the ERK signaling pathway was examined. In melatonin-treated cells, the expression of p-ERK1/2 increased significantly (*p* < 0.05; Figure 6A,B). When the cells were treated with PD98059 and melatonin, INHA expression decreased significantly (*p* < 0.05; Figure 6C,D). Next, we tested the proliferation of chicken Sertoli cells after inhibiting the ERK signaling pathway. The results showed that PD98059 (ERK1/2 inhibitor) could reverse the positive effects of melatonin on cell viability (Figure 6E) and proliferation (Figure 6F,G). The addition of the ERK inhibitor PD98059 significantly reduced the levels of CCND1 (Figure 6I,J,K) and PCNA (Figure 6H,J,L) compared with the melatonin-only group (*p* < 0.05). These results demonstrated that melatonin promotes Sertoli cell proliferation by activating the ERK signaling pathway and affecting INHA.

## 3. Discussion

The proliferation of testicular Sertoli cells affects the spermatogenic function of the testes, determining the final volume of the testes and sperm production [26]. However, the effects of melatonin on the proliferation of chicken Sertoli cells have never been studied. Our results show that melatonin could promote the proliferation of chicken Sertoli cells by activating the ERK/inhibin alpha subunit signaling pathway.

We found that the 1000 nM melatonin treatment can significantly promote cell proliferation and increase the expression of proliferation-related genes and proteins (PCNA, CCND1). Cell proliferation nuclear antigen (PCNA) regulates cell proliferation and it is mainly expressed in the S phase [27]. CCND1 is a regulatory subunit of the cyclin-dependent kinases CDK4 and CDK6 required for the G0/G1 to S phase transition [28]. Our results are consistent with previous studies. For example, melatonin has a proliferative effect in a variety of cells and in mouse spermatogonia, melatonin not only promotes cell proliferation by the extracellular signal-regulated kinase 1/2 (ERK1/2) pathway, but also by increasing metallothionein 2 (Mt2) [29]. In mesenchymal stem cells, melatonin promotes cell proliferation by inducing expression of the SRY-box transcription factor 2 gene and preventing replicative senescence [30]. Similarly, a melatonin treatment of neural stem cells in hypoxia is a powerful strategy which could increase cell proliferation and reduce cell death [31].

Inhibin plays an important role in cell proliferation. In this study, melatonin promoted Sertoli cell proliferation and significantly increased INHA expression. We hypothesized that melatonin affected Sertoli cell proliferation through INHA. Subsequently, the silencing of INHA reversed the melatonin-induced proliferation and the up regulation of the proliferation-related genes and proteins (PCNA and CCND1). These results indicated that melatonin promotes Sertoli cell proliferation via INHA. Once again, our results are consistent with prior work. In bovine Sertoli cells, melatonin promotes INHA gene expression [16]. In mouse testes, α-solanine inhibits cell proliferation by reducing mitochondrial function and INHA synthesis [26]. The expression of CCND1 and CCNE was also significantly reduced after silencing INHA in mouse testis Sertoli cells [25]. In pig granulosa cells, INHA promotes the expression of PCNA and CCNB1 and inhibits the expression of Caspase-3 and BAX. After interference with INHA, gene expression was reversed. The results indicated that INHA has pro-proliferative effects [24]. Together, these studies support a role for melatonin promotes cell proliferation via INHA.

Mitogen-activated protein kinase (MAPK) is an important mediator in signaling pathways. There are three subtypes of the mitogen-activated protein kinase pathway: extracellular-regulated protein kinase (ERK), c-Jun NH2-terminal kinase (JNK), and p38 MAPK. ERK mainly regulates cell proliferation, metastasis, and differentiation [32]. Our results indicated that melatonin significantly increased the level of p-ERK1/2 and the inhibition of the ERK1/2 signaling pathway reversed melatonin-induced Sertoli cell proliferation and down-regulated the expression of CCND1, PCNA, and INHA. Recently study has shown that melatonin promotes viability in human nucleus pulposus cells (NPC) and activates the ERK signaling pathway, but inhibition of the ERK signaling pathway reverses the effects of melatonin on NPC [33]. Melatonin can also promote osteoblast differentiation through the ERK signaling pathway [34]. Moreover, melatonin stimulates the production of glial cell line-derived neurotrophic factor (GDNF) in Sertoli cells by activating the ERK1/2 signaling pathway and promotes the proliferation of goat spermatogonial stem cells (SSC) [19]. These studies support our results and showing that melatonin promotes cell proliferation through the ERK signaling pathway. However, in vivo experiments are required to confirm the effects of melatonin on male fertility.

Melatonin has been reported to be associated with male reproduction. Men with oligozoospermia, asthenospermia, or non-obstructive azoospermia have significantly lower melatonin levels compared to fertile men [35]. In mice, melatonin pretreatment can alleviate heat-induced testicular cell apoptosis and oxidative stress, and post-treatment of melatonin may promote testicular recovery from heat-induced injury by maintaining the integrity of supportive cell tight junctions [36]. Our findings provide more information on the regulatory mechanism of proliferation of Sertoli cells which would lay the foundation for developing new technologies for the regulation of male reproduction.

## 4. Materials and Methods

This study was performed at Jilin Agricultural University and was approved by the Ethics Committee of Jilin Agricultural University, China.

### 4.1. Primary Sertoli Cell Culture

Testicles were extracted from 3 to 5-week old chickens and cleaned 3 times with phosphate-buffered saline (PBS) containing penicillin and streptomycin, following the removal of the testicular tunica and blood vessels. The stripped testicles were transferred to a beaker and immersed several times in PBS, penicillin, and streptomycin. The testicles were cut into small pieces (1 mm^3^) with scissors, placed in a 10 mL centrifuge tube, and centrifuged to pellet the sample. Five milliliters of collagenase IV (1 mg/L; Solarbio, Beijing, China) was added to the centrifuge tube and the testicular tissues were digested at 37 °C in a CO_2_ incubator for 40–50 min. The collagen-digested tissues were pelleted (via centrifugation) and then digested with 10 volumes of 0.25% trypsin-EDTA (Solarbio) for 6–8 min (the pipette was continuously blown during the digestion). Five milliliters of PBS was added to the centrifuge tube before centrifuging at 800 rpm for 1 min. The supernatant was filtered through a 300 mesh sieve into a Petri dish, and the fetal bovine serum was added to terminate the digestion. The remaining tissue pieces were further digested with a trypsin-EDTA digestion solution for 6–8 min, filtered through a 300 mesh sieve, and added to the serum-containing medium to terminate digestion. The tissue pieces were repeatedly digested until complete digestion. After digestion termination, all of the filtrates were mixed into a new centrifuge tube and centrifuged at 1000 rpm for 5 min to collect the cells. 

The collected cells were cultured in a DMEM/F12 medium containing 10% FBS, penicillin, and streptomycin at 37 °C in a 5% CO_2_ incubator for 24 h. The cells were then removed from the culture solution and cleaned twice with PBS. Next, the cells were hypotonic treated (PBS: distilled water = 1: 2) for 3 min and again cleaned with PBS (3 times). Finally, the cells were cultured in new culture dishes with fresh culture medium (DMEM/F12 medium containing 10% FBS, penicillin, and streptomycin).

### 4.2. Sertoli Cell Treatment

To identify the appropriate concentration of melatonin, chicken Sertoli cells were treated with 1, 10, 100, and 1000 nM melatonin (Sigma, Saint Louis, MO, USA) for 48 h (Time point of pre-laboratory screening). The cells were pretreated with an ERK inhibitor PD98059 (20 μM, MedChemExpress, Monmouth Junction, NJ, USA) for 2 h to study the action of the extracellular-regulated protein kinase (ERK) signaling pathway. Each experiment in this article was repeated 3 times, that is n = 3.

### 4.3. Oil Red O Staining

After 24 h of in vitro cell culture (2 × 10^6^ cells/medium dishe), the culture solution was discarded, and the cells were set with 4% PFA-PBS (paraformaldehyde; Huacheng, Changchun, China) at 4 °C for 1 h and cleaned twice with PBS. The cells were then dyed with a freshly diluted oil red O (Solarbio) filtrate at room temperature for 30 min. The dye solution was discarded, separated by 60% ethanol for 5 s, and washed with water. The cells were counterstained with hematoxylin (NOVON, Beijing, China) for 30 s, washed with water until blue, and immediately observed and imaged using an optical microscope (Motic-BA410E, Xiamen, China).

### 4.4. PCR and Agarose Gel Electrophoresis

Total RNA was obtained from the Sertoli cells using the Trizol method. cDNA was generated from 1 ng of RNA using a reverse transcription kit (TaKaRa, Tokyo, Japan), per the manufacturer’s instructions. The cDNA was subjected to PCR using the primers in Table 1. The PCR product was electrophoresed on a 1 % agarose gel (BIOWEST, Beijing, China) and imaged (Tanon-2005R, Shanghai, China).

### 4.5. Immunofluorescence Assay

Sertoli cells were sowed at 1 × 10^5^ cells/well in 24-well plates. The cells were fixed on cell slides with 4% PFA, washed 3 times, and permeabilized with 1% Triton X-100 (Solarbio). After 3 washes, the cells were blocked with 1% BSA (AMRESCO, Solon, Ohio, USA), washed three times, and then incubated at 4 °C overnight with a rabbit anti-vimentin primary antibody (Sigma). The next day, the cells were washed 6 times and incubated with a goat anti-rabbit antibody in darkness. Next, the cells were washed 6 times and incubated with Hoechst 33258 (Beyotime Biotechnology, Shanghai, China) in darkness. Finally, the cells were washed 6 times and, the slides were fixed and observed under a full-functional cell imaging detector (BioTek, Winooski, VT, USA). The test was washed with PBS.

### 4.6. CCK-8 Assay

A Cell Counting Kit 8 (CCK-8) was used to measure cell viability. Sertoli cells were sowed at 2 × 10^4^ cells/well in 96-well plates. The cells were treated with melatonin for 48 h, followed by 12 μL of CCK-8 (Bimake, Houston, TX USA). After 2 h, the absorbance was measured at 450 nm using a microplate reader (BioTek).

### 4.7. Cell Transfection

Sertoli cells were sowed at 2 × 10^6^ cells/well in medium dishes. The INHA interference sequence (Table 2) was transiently transfected into Sertoli cells using Lipofectamine^TM^ 2000 (70% confluence of cells). After transfection, the expression of INHA in SC was measured using real-time qPCR and an enzyme-linked immunosorbent assay (ELISA). Following verification that the interference was successful, the cells were treated with 1000 nM melatonin for 48 h.

### 4.8. Total RNA Isolation and Real-Time qPCR

Total RNA was obtained as described above (see Section 4.4). Real-time qPCR was performed on a StrataGene Mx3005P system (Agilent, Santa Clara, CA, USA) using 2×RealStar Green Fast Mixture with ROX II (GenStar, Beijing, China) and the primers shown in Table 3. The relative expression fraction of the target gene was computed by means of the 2^-ΔΔCt^ method and expressed in comparison with the GAPDH gene.

### 4.9. EdU Staining

The BeyoClick^TM^ EdU kit (Beyotime Biotechnology) was used to examine the effects of melatonin or silencing INHA on Sertoli cells. Sertoli cells were sowed at 1 × 10^5^ cells/well in a 24-well plate and processed according to experimental requirements (see Section 4.2). The production of fluorescent slides was performed according to the manufacturer’s instructions, the nuclear staining reagent is Hoechst 33342 (BeyoClick^TM^ EdU kit, Beyotime Biotechnology, Shanghai, China). The proportions of EdU-positive cells were observed using a full-function cell imager detector (BioTek) and calculated with the Gene5 software.

### 4.10. Enzyme-Linked Immunosorbent Assay (ELISA)

The cell supernatant was processed according to the ELISA kit instructions (Shanghai Mlbio Biotech, Shanghai, China). The INHA level was measured within 15 min at 450 nm using a microplate reader.

### 4.11. Western Blotting Assay

Sertoli cells were sowed at 2 × 10^6^ cells/well in medium dishes and processed according to experimental requirements. Sertoli cells were digested in a centrifuge tube, rinsed once with PBS, and an appropriate protein lysis buffer (RIPA, Beyotime) was added (dependent on cell volume). The supernatant was collected by centrifugation at 4 °C. The protein concentration was measured using a BCA kit (Beyotime), and the protein was boiled 5 times in protein loading buffer for 10 min. A total of 60 μg of protein was isolated by SDS-PAGE and transferred to a PVDF membrane (Merck Millipore, Darmstadt, Germany). The membrane was incubated with a blocking buffer (LI-COR Biosciences, Lincoln, NE, USA) for 1 h and then incubated with a primary antibody at 4 °C overnight. The membrane was washed 5 times with TBST buffer (Sangon Biotech, Shanghai, China), incubated with the second antibody for 1 h at room temperature, and then washed 5 times with TBST. The protein bands were analyzed on a chemical isotope imaging system (CLiNX Scientific Instruments, Shanghai, China). The antibody information is provided in Table 4.

### 4.12. Statistical Analysis

Graph Pad Prism 5.0 software (GraphPad Prism Software, San Diego, CA, USA) was used to analyze the experimental data. The results are expressed as mean ± SEM, and the differences between the groups were tested by one-way analysis of variance (ANOVA) and *t*-tests. The results were considered statistically significant at *p* < 0.05.

## 5. Conclusions

In conclusion, our results demonstrate that melatonin promotes cell proliferation and the upregulation of mRNA and protein expression levels of CCND1 and PCNA in chicken Sertoli cells via activation of the ERK/inhibin alpha subunit signaling pathway. 

## Figures and Tables

**Figure 1 molecules-25-01230-f001:**
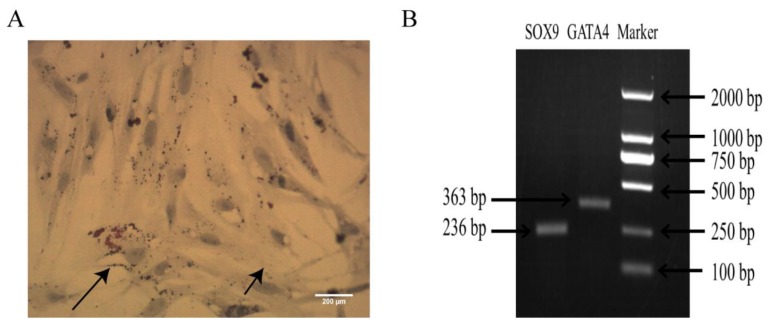
Identification of Sertoli cells. (**A**) The short arrow indicates bipolar bodies and the long arrow indicates lipid droplets, at ×40 magnification. (**B**) Expression of SOX9 and GATA4 mRNA. (**C**) The nucleus was counterstained with Hoechst 33258 and green fluorescence indicates vimentin positive (×4 magnification). N = 3 for both.

**Figure 2 molecules-25-01230-f002:**
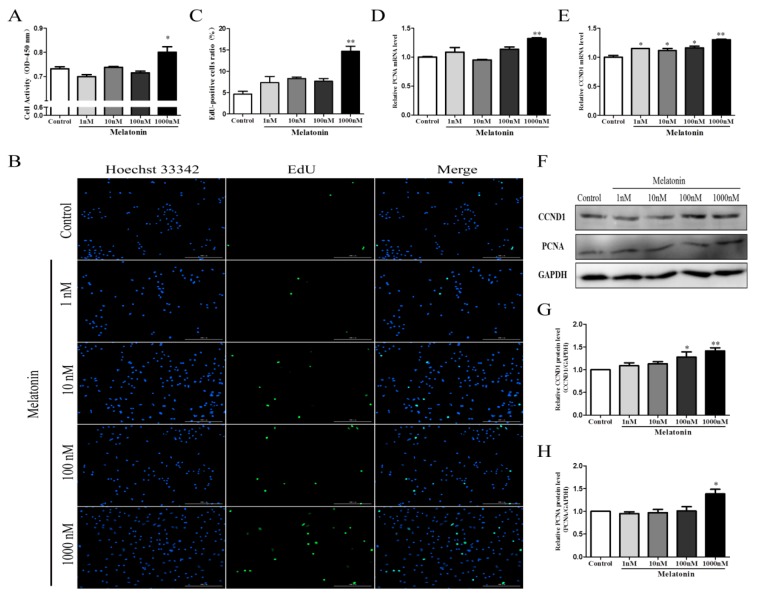
Effects of melatonin on the proliferation of chicken Sertoli cells. (**A**) Cell activity of chicken Sertoli cells (n = 3). (**B**) The EdU (5-ethynyl-2’-deoxyuridine) method was used to measure chicken Sertoli cell proliferation (×10 magnification; n = 3). (**C**) Statistical analysis of data in (**B**). The relative mRNA expression levels of (**D**) proliferating cell nuclear antigen (PCNA) and (**E**) Cyclin D1 (CCND1; n = 3 for both). (**F**) The relative protein expression levels of CCND and PCNA. Quantitative analyses of the (**G**) CCND1 and (**H**) PCNA protein results (n = 3 for both). ** *p* < 0.01; * *p* < 0.05.

**Figure 3 molecules-25-01230-f003:**
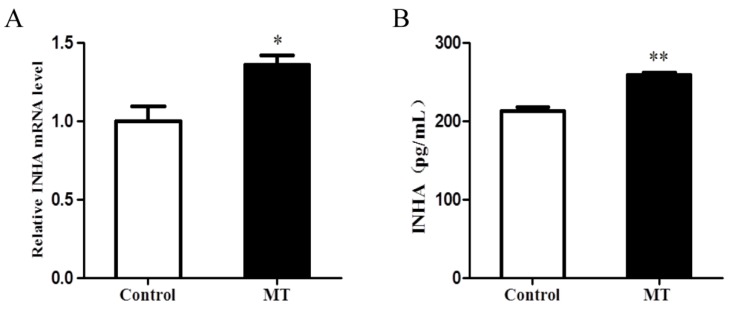
Effects of melatonin (1000 nM) on the INHA expression of chicken Sertoli cells. (**A**) Relative mRNA expression levels of INHA and (**B**) INHA measured by ELISA (n = 3). ** *p* < 0.01; * *p* < 0.05.

**Figure 4 molecules-25-01230-f004:**
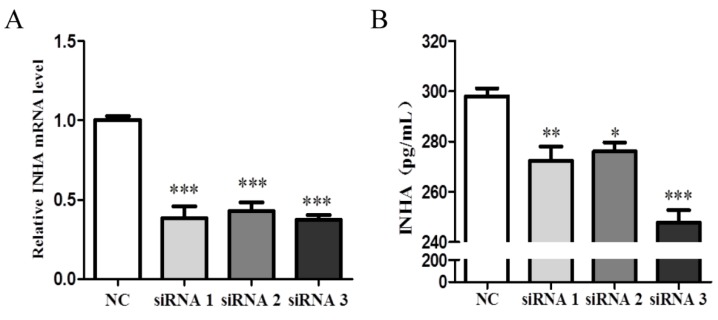
The interference efficiency of INHA siRNA. (**A**) Cells were treated with a negative control (NC) siRNA or INHA siRNA. After 24 h, RT-qPCR was used to measure INHA mRNA expression (n = 3). (**B**) ELISA was also used to measure INHA levels (n = 3). *** *p* < 0.001; ** *p* < 0.01; * *p* < 0.05.

**Figure 5 molecules-25-01230-f005:**
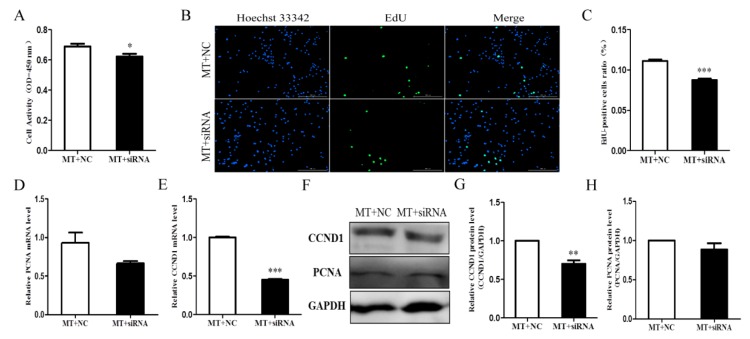
Effects of melatonin on Sertoli cell proliferation after silencing INHA. (**A**) Cell activity of chicken Sertoli cells (n = 3). (**B**) The EdU method was used to measure chicken Sertoli cell proliferation (×10 magnification; n = 3). (**C**) Statistical analysis of data in (**B**). The relative mRNA expression levels of (**D**) proliferating cell nuclear antigen (PCNA) and (**E**) Cyclin D1 (CCND1; n = 3 for both). (**F**) The relative protein expression levels of CCND1 and PCNA. Quantitative analyses of the (**G**) CCND1 and (**H**) PCNA protein results (n = 3 for both). *** *p* < 0.001; ** *p* < 0.01; * *p* < 0.05.

**Figure 6 molecules-25-01230-f006:**
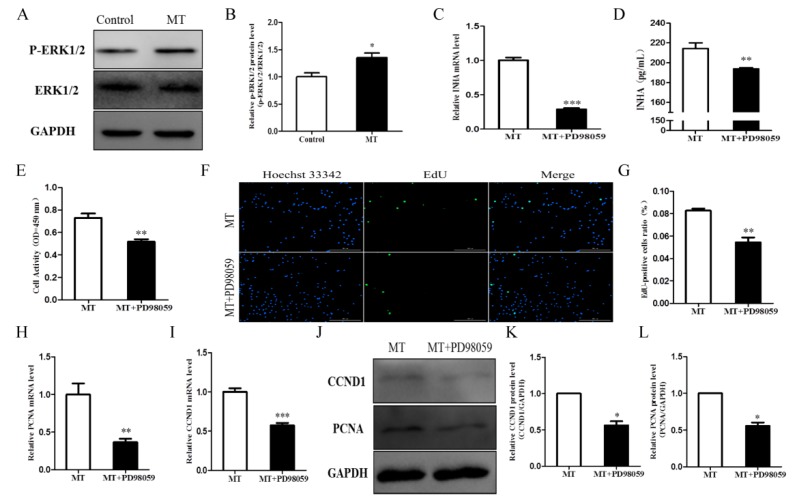
Effects of melatonin and PD98059 on cell proliferation and INHA expression in chicken Sertoli cells. (**A**) The protein level of p-ERK1/2 (n = 3). (**B**) Quantitative analyses of the p-ERK protein results. (**C**) INHA mRNA levels of chicken Sertoli cells (n = 3). (**D**) INHA levels of chicken Sertoli cells by ELISA (n = 3). (**E**) Cell viability of chicken Sertoli cells (n = 3). (**F**) The EdU method was used to measure chicken Sertoli cell proliferation (×10 magnification; n = 3). (**G**) Statistical analysis of data in (**F**). The relative mRNA expression levels of (**H**) proliferating cell nuclear antigen (PCNA) and (**I**) Cyclin D1 (CCND1; n = 3 for both). (**J**) The relative protein expression levels of CCND1 and PCNA. Quantitative analyses of the (**K**) CCND1 and (**L**) PCNA protein results (n = 3 for both). *** *p* < 0.001, ** *p* < 0.01 or * *p* < 0.05.

**Table 1 molecules-25-01230-t001:** Gene primer sequence.

Genes	Primer Sequence (5′–3′)	Genebank No.	Size (bp)
SOX9	F: GCTGTGGAGGCTGCTGAATGAG	NM_204281.1	236
	R: CGCTGATGCTGGAGGATGACTG		
GATA4	F: TGTCACCTCGCTTCTCCTTCTCC	XM_004935896.3	363
	R: AGTGCCCTGTGCCATCTCTCC		

**Table 2 molecules-25-01230-t002:** INHA siRNA sequences.

siRNA	Sequence (5′→3′)
Negative control	Sense: UUCUCCGAACGUGUCACGUTT Antisense: ACGUGACACGUUCGGAGAATT
INHA siRNA1	Sense: GCGUCCCUCAACAUCUCUUTT Antisense: AAGAGAUGUUGAGGGACGCTT
INHA siRNA2	Sense: CCACGGGAACUGUGCCGAATT Antisense: UUCGGCACAGUUCCCGUGGTT
INHA siRNA3	Sense: ACCUCUGAUGGUGGCUACUTT Antisense: AGUAGCCACCAUCAGAGGUTT

**Table 3 molecules-25-01230-t003:** Primer sequences for real-time quantitative PCR.

Genes	Primer Sequence (5′–3′)	Genebank No.	Size (bp)
PCNA	F: GCAGATGTTCCTCTCGTTGTGGAG	NM_204170.2	95
	R: GAGCCTTCCTGCTGGTCTTCAATC		
CCND1	F: TCGGTGTCCTACTTCAAGTG	NM_205381.1	273
	R: GGAGTTGTCGGTGTAAATGC		
INHA	F: ACCGCAGAGATGTCCTCGAAGAG	NM_001031257.1	95
	R: GCACGGCACGTCTGTGGAAG		
GAPDH	F: TAAGCGTGTTATCATCTC	NM_204305.1	83
	R: GGGACTTGTCATATTTCT		

**Table 4 molecules-25-01230-t004:** The information of antibodies.

Antibodies	Cat NO.	Source	Dilution
PCNA	bs-2006R	Bioss, Beijing, China	1:300
CCND1	PAB9944	Abnova, Taipei, Taiwan, China	1:300
GAPDH	60004-1-lg	ProteinTech, Chicago, IL, USA	1:10,000
ERK1/2	orb315598	Biorbyt, California, USA	1:1000
p-ERK1/2	orb338969	Biorbyt, California, USA	1:1000
Goat Anti-Rabbit IgG	SA00001-2	ProteinTech, Chicago, IL, USA	1:10,000
Goat Anti-mouse IgG	SA00001-1	ProteinTech, Chicago, IL, USA	1:10,000

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
