# Peer review of "Melatonin Promotes the Proliferation of Chicken Sertoli Cells by Activating the ERK/Inhibin Alpha Subunit Signaling Pathway"

_molecules, 2020, doi:10.3390/molecules25051230_

Round 1

Reviewer 1 Report

In their study entitled “Melatonin promotes the proliferation of chicken Sertoli cells by activating the ERK/inhibin alpha subunit signaling pathway” authors have investigated the effects of melatonin on cell proliferation and its underlying mechanisms in chicken Stertoli cells. Their results showed that 1000 nM melatonin promoted cell proliferation in chicken Sertoli cells, and this was done by activating the ERK/inhibin alpha subunit signaling pathway.

Although this is an interesting study and authors pointed that, the effects and mechanisms of melatonin on Sertoli cell proliferation have not been studied before; there are some additional points that authors have to address.

I advise the authors to write in the introduction part more detailed about melatonin, as it is well known that melatonin have pleiotropic functions. For example, melatonin is involved in circadian regulation. Melatonin is also a potent antioxidant and acts as a free radical scavenger and as such, melatonin can serve as an anti-aging molecule. Melatonin retains anti-apoptotic and anti-tumour properties as well. Also, to write briefly about arylalkylamine N-acetyltransferase (AANAT) enzyme as critical regulatory element of melatonin synthesis.

In the Discussion section, I recommend authors to comment some in vivo experiments on melatonin promoting sell proliferation of Sertoli cells (they already mentioned that in vivo experiments are required) so it is interesting to compare known results from in vitro and in vivo experiments on other animals, if they are available. Also to write a bit more on possible effect of melatonin on male fertility. Add future directions to the discussion. How can researchers use the information in this study? This will improve the relevance of results obtained in this study.

Reviewer 2 Report

Overview:

The present study is an in vivo evaluation of melatonin on the proliferation of chicken Sertoli cells. These results show that melatonin promote the proliferation of chicken Sertoli cells by activating the ERK/inhibin alpha subunit signaling pathway.

The protective role of melatonin on male reproduction is well known, e.g.:

  1. Zhang P, Zheng Y, Lv Y, et al. Melatonin protects the mouse testis against heat-induced damage [published online ahead of print, 2020 Jan 16]. Mol Hum Reprod. 2020;gaaa002. doi:10.1093/molehr/gaaa002
  2. Niu B, Li B, Wu C, et al. Melatonin promotes goat spermatogonia stem cells (SSCs) proliferation by stimulating glial cell line-derived neurotrophic factor (GDNF) production in Sertoli cells. Oncotarget. 2016;7(47):77532–77542. doi:10.18632/oncotarget.12720
  3. Yu K, Deng SL, Sun TC, Li YY, Liu YX. Melatonin Regulates the Synthesis of Steroid Hormones on Male Reproduction: A Review. Molecules. 2018;23(2):447. Published 2018 Feb 17. doi:10.3390/molecules23020447
  4. Rossi SP, Windschuettl S, Matzkin ME, et al. Melatonin in testes of infertile men: evidence for anti-proliferative and anti-oxidant effects on local macrophage and mast cell populations. Andrology. 2014;2(3):436–449. doi:10.1111/j.2047-2927.2014.00207.x

But the effects of melatonin on chicken Sertoli cells are unknown. It would be appreciated if the Authors expanded the introduction and discussion to the current knowledge about the effect of melatonin on the male reproductive system

I have too many questions to be able to recommend this paper for publication as it is now. Therefore, I recommend that a major revision is warranted.

Major comments:

  1. Since there is a paper Niu B, Li B, Wu C, et al. Melatonin promotes goat spermatogonia stem cells (SSCs) proliferation by stimulating glial cell line-derived neurotrophic factor (GDNF) production in Sertoli cells. Oncotarget. 2016;7(47):77532–77542. doi:10.18632/oncotarget.12720 the sentence “However, the effects and mechanisms of melatonin on Sertoli cell proliferation have not been studied” (page 2 lines 44-45) is not truth. Please correct it.
  2. What was the density of the cell culture?
  3. Why the Authors use only 48 h treatment?
  4. What does it mean ‘fresh culture medium’ (p. 7 lines 202-203). Please describe all components of the medium.
  5. What the Authors mean saying ‘hypotonic treated’?
  6. The ‘n’ value is low – only 3. Please describe the replication number also. What the Authors mean saying ‘n’ value?
  7. Why the GAPDH gene was chosen as reference?

Minor comments:

  1. There should be scientific hypothesis in the Introduction.
  2. There is no information about the ‘n’ in Fig. 1.
  3. The Fig. 2B description is wrongly written. There is no information about Hoechst 33342 staining.
  4. How much ng of RNA was used to generate cDNA?
  5. The description about immunofluorescence assay is not correct. There is wrong Hoescht number and there is no information about EdU.

Round 2

Reviewer 2 Report

All my questions /comments have been clarified by the Authors. In my opinion, the manuscript is ready for publication.